# Why think step by step? Reasoning emerges from the locality of experience

**Ben Prystawski**
Department of Psychology
Stanford University
Stanford, CA 94305
benpry@stanford.edu

**Michael Y. Li**
Department of Computer Science
Stanford University
Stanford, CA 94305
michaelyli@stanford.edu

**Noah D. Goodman**
Departments of Psychology and Computer Science
Stanford University
Stanford, CA 94305
ngoodman@stanford.edu

## Abstract

Humans have a powerful and mysterious capacity to reason. Working through a set of mental steps enables us to make inferences we would not be capable of making directly even though we get no additional data from the world. Similarly, when large language models generate intermediate steps (a chain of thought) before answering a question, they often produce better answers than they would directly. We investigate why and how chain-of-thought reasoning is useful in language models, testing the hypothesis that reasoning is effective when training data consists of overlapping local clusters of variables that influence each other strongly. These training conditions enable the chaining of accurate local inferences to estimate relationships between variables that were not seen together in training. We prove that there will exist a "reasoning gap", where reasoning through intermediate variables reduces bias, for the simple case of an autoregressive density estimator trained on local samples from a chain-structured probabilistic model. We then test our hypothesis experimentally in more complex models, training an autoregressive language model on samples from Bayes nets but only including a subset of variables in each sample. We test language models' ability to match conditional probabilities with and without intermediate reasoning steps, finding that intermediate steps are only helpful when the training data is locally structured with respect to dependencies between variables. The combination of locally structured observations and reasoning is much more data-efficient than training on all variables. Our results illustrate how the effectiveness of reasoning step by step is rooted in the local statistical structure of the training data.

## 1 Introduction

The human mind is a ship – the immediate inferences we make from instinct keep us afloat, but reason is the compass that brings us to the shore of wisdom. Many tasks that we find hard to do immediately – solving math problems, planning vacations, understanding our relatives – become much easier when we talk ourselves through a reflective reasoning process. Likewise, by considering thought experiments or "intuition pumps" in science we can form strong beliefs – such as that the rate at which an object falls should not depend on its mass – purely by thinking through a set of steps

[1, 2]. It is not *a priori* obvious that step-by-step reasoning should be helpful. Reasoning does not give us any new data from the world, yet it can still improve our inferences. In investigating the origins of reasoning, we must thus ask, why does reasoning help at all?

Large language models have been shown capable of performing a wide variety of tasks by immediately answering a question [3–6]. However, they struggle with some complex tasks like math word problems [7]. A recent line of work has demonstrated that inducing language models to produce a "chain of thought" consisting of intermediate steps toward a solution, before giving an answer, leads to better performance than prompting them to produce an answer directly [8–11]. Other work has built on these findings, showing that providing worked solutions in context is helpful across a broad array of tasks [12, 13] and proposing methods to augment language models' reasoning performance [14–16]. These findings raise an important question: *why* is chain-of-thought reasoning useful? Exploring this question in language models may also provide insight into the origins of human reasoning.

We hypothesize that chain-of-thought reasoning is useful in language models due to *local* structure in the training data. Human experience is governed by our first-person perspective: we see and hear aspects of the world that are near to us in time and space. Yet our reasoning transcends these local bounds, supporting plans and conclusions that span distance and years. Similarly, language models are trained on documents in natural language, which are usually about a few closely interconnected topics [17, 18]. When concepts co-occur frequently in experience or training data, estimating the effect of one on the other is easy to do directly with simple statistical estimators. However, when we need to infer the effect of one piece of information on another but have not encountered them together, we must make a series of inferences that jump between pairs of concepts to connect what we know with what we want to infer. We posit that chain-of-thought reasoning becomes useful exactly when the training data is structured locally, in the sense that observations tend to occur in partially overlapping neighborhoods of related concepts.

To illustrate, we may know the value of some variable $A$ and want to know about another variable $C$, so we try to estimate $P(C|A)$. However, if we need to estimate probabilities using observed samples from joint distributions and we have not often seen $A$ and $C$ together, we would struggle to estimate $P(C|A)$ directly. Instead, we might estimate it by reasoning through intermediate variables. If conditioning on an intermediate variable $B$ renders $A$ and $C$ independent of each other, we can compute the conditional probability by marginalizing over $B$, using the fact that $P(C|A) = \sum_B P(C|B)P(B|A)$. For an example in natural language, suppose that we want to answer a question like "What is the climate of France's capital?" using a model trained on text from Wikipedia. Wikipedia pages for cities usually have information about the city's climate and Wikipedia pages for countries have information about capital cities, but pages for countries typically do not directly mention the climate of the capital city. It is possible that our model could succeed in answering the question directly, but it would likely have more success if it took the intermediate step of stating the capital of France. By introducing the step "the capital of France is Paris" before answering "Paris has an oceanic climate," an autoregressive model can better leverage the dependencies that are well-represented in its training data.

Our hypothesis is similar to that advanced by Chan et al. [19] to explain why in-context learning occurs (generalizing from examples shown in context). They show that "burstiness," the tendency of instances of a class to occur close together, in the training data is important for language models to learn to infer tasks based on examples in the context window. Burstiness and locality are two different statistical properties of training data that could arise from topic structures in language models or from a first-person perspective in humans. While burstiness concerns how a class is distributed over time, locality concerns which classes co-occur together. In datasets with local structure, concepts that strongly influence each other are also seen together frequently in the training set, whereas concepts that do not are seen together less frequently.

In this work, we investigate the properties of a model's training data that make generating intermediate variables useful for inference. We first formalize the problem mathematically and prove that reasoning through the right intermediate variables reduces bias in a model trained on local samples from a chain. We then train autoregressive transformers from scratch on synthetic data with different structures and evaluate the models' ability to match conditional probabilities for pairs of variables that were not observed together in training. Generating values for relevant intermediate variables enables models to match conditional probabilities more accurately, but only when the training data is locally structured. Allowing the model to select intermediate variables to reason through is similarly helpful to giving it

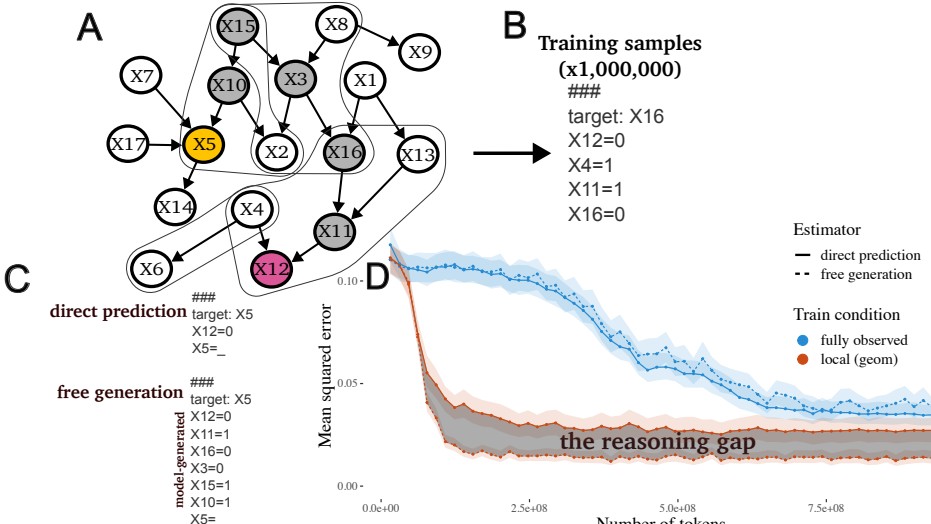

Figure 1: Overview of our training and estimation setup. A: visualization of a Bayes net. The pink variable is an example observed variable and the yellow variable is an example target variable. Gray variables are examples of useful intermediate variables for reasoning. Lines show examples of local neighborhoods from which training samples are drawn. B: format of the training samples. C: illustration of direct prediction and free generation estimators as prompts. We prompt the model to either immediately predict the target variable (direct prediction) or do so after generating intermediate variables and their values (free generation). We then compute mean squared errors between the estimated and true conditional probabilities. D: mean squared error by number of training tokens for each training condition and estimator. Ribbons indicate 95% confidence intervals.

the ideal intermediate variables to reason through. Our results suggest that chain-of-thought reasoning is useful for language models because 1) direct prediction is inaccurate for some inferences because the relevant variables are rarely seen together in training and 2) chain-of-thought reasoning improves estimation by incrementally chaining local statistical dependencies that are observed frequently in training. We also find that the combination of locally structured training data and reasoning with self-generated intermediate variables yields much greater data efficiency than training on data containing all variables.

## 2 Task setup

Our theory and experiments take place in the context of conditional inference in joint distributions, as represented by Bayesian networks. In our framing, a learner sees samples from a Bayes net and needs to accurately estimate conditional probabilities. However, the learner may not see all variables together, having access only to locally-structured observations instead.

We describe our framework formally. First, we assume there is some underlying Bayes net with random variables $\{Y_i\}_{i=1}^N$ taking support on a finite set $\mathcal{X}$. We denote the distribution defined by the Bayes net as $p_d$, the *data distribution*. Our training data is a sequence of *variable indices* $i \in \{1, \ldots N\}$, and *variable values* $v_i \in \mathcal{X}$. Variable values always follow variable indices.

### 2.1 Observation distribution

We create local structure in our task via an *observation distribution* $p_{obs}$, which is a distribution over subsets of the variable indices. Observation distributions take support on a set $\mathcal{Y}_{obs} \subseteq \mathcal{P}(\{1, \ldots, N\})$. As a consequence, our learner will only see samples from the joint distribution over certain subsets of random variables in the Bayes net. The generative process for our sequence consists of two steps. First, we sample a set of variable indices $\{i_t\}_{t=1}^K$ according to $p_{obs}$. These indices correspond to some set of random variables $\{Y_{i_t}\}_{t=1}^K$ in the Bayes net, which appear in random order in training. We

then sample the values $\{v_{i_t}\}_{t=1}^K$ corresponding to the variable indices according to the distribution $p_d(Y_{i_1}, \ldots, Y_{i_K})$. Together, $p_d$ and $p_{\text{obs}}$ define a distribution $p$ on training sequences.

We are particularly interested in local observation distributions as they enable models to learn strong dependencies and reflect some of the statistical structure of human experience. In local observation distributions, the learner observes a subset of variables that are close together in the Bayes net graph. Figure 1A shows several local neighborhoods in an example Bayes net.

## 2.2  Estimators

Given an autoregressive conditional probability estimation model $q$ that is trained to predict variable indices and values, we can use one of three different estimators for the conditional probabilities. The estimators differ in how a sequence of variables and values is produced before estimating the target variable. If we want to compute the conditional probability of a target variable $Y_i$ taking some value $y_i$ given that we have observed variable $Y_j$ to be $y_j$, we can use one of the following estimators.

**Direct prediction**   The direct prediction estimator is the baseline without any reasoning: the probability directly output by $q$.

$$\hat{q}_D(Y_i = y_i | Y_j = y_j) = q(Y_i = y_i | Y_j = y_j). \tag{1}$$

We simply use the model to directly estimate the conditional probability of the target variable.

**Scaffolded generation**   The scaffolded generation estimator represents ideal reasoning if we know the best set of steps to work through. A scaffold is an ordered set $S$ consisting of variables that were each observed together with the observed or target variable (or another scaffold variable) and collectively $d$-separate the observed variable from the target variable. In the case of a chain, the scaffold consists of all variables between $Y_i$ and $Y_j$. Variables are ordered by their distance from the observed variable in the Bayes net, increasing. We estimate each variable given the observed variable and previously generated scaffold variables using $q$ before estimating the target probability. We approximately marginalize over the scaffold variables' values using $M$ Monte Carlo samples from the conditional distributions.

$$\hat{q}_S(Y_i = y_i | Y_j = y_j) = \frac{1}{M} \sum_{k=1}^M q(Y_i = y_i | \{Y_s = y_s^{(k)}\}_{s \in S}, Y_j = y_j) \tag{2}$$

$$\text{where } y_s^{(k)} \sim q(Y_s | \{Y_t = y_t^{(k)}\}_{t \in S | t \prec s}, Y_j = y_j) \tag{3}$$

We will denote the distribution defined by the sampling procedure in 2 by $\hat{q}_S$.

**Free generation**   The free generation estimator tests whether trained models spontaneously generate useful intermediate variables. It is similar to scaffolded generation in that it generates intermediate variables before the target, but free generation uses the model to choose *which* variables to instantiate in addition to estimating their values. We simply sample variable indices and values from $q$ until it generates the index of the target variable. At that point, we compute the probability of the target variable. We average over $M$ such samples.

## 3  Theoretical results

In order to understand the range of situations in which reasoning might be useful, we study the conditions under which a risk-minimizing sequence model $q$ benefits from reasoning through intermediate variables. We analyze a simplified version of our task in which a model is trained on pairs of adjacent variables from a directed chain. In this setting, we prove that the minimizer of a risk consisting of cross entropy with entropy regularization exhibits a *reasoning gap*: the bias of direct conditional probability estimates between pairs of random variables that do not appear together is higher than indirect estimates that chain together conditional probabilities of intermediate random variables.

We assume that a risk minimizer $q$ is trained on a sequence of alternating *variable indices* $i_t \in \{1, \ldots, N\}$ and *variable values* $Y_i \in \mathcal{X}$ where the $Y_i$'s take support in some finite set $\mathcal{X}$. Variable values always follow variable indices. We assume that the joint distribution $p_d$ over $Y_1, Y_2, \ldots, Y_N$

factorizes as $p_d(Y_1, \ldots, Y_N) = p_d(Y_1) \prod_{i=1}^{N} p_d(Y_{i+1}|Y_i)$. The observation distribution $p_{\text{obs}}$ only assigns non-zero probability to adjacent variable pairs, i.e. $p_{\text{obs}}(\{i,j\}) = 0$ if $|i - j| > 1$ and $p_{\text{obs}}(X) = 0$ if $|X| \neq 2$. The training set consists of i.i.d. samples from $p$, which is the distribution over complete sequences of variable indices and values defined by $p_{obs}$ and $p_d$.

We show that when our data has the locality structure as described, marginalizing over intermediate random variables using the learned conditional probabilities between adjacent variables leads to lower-bias estimates than using the learned estimate directly:

**Theorem 3.1.** *Let $\mathcal{S}$ be the space of possible sequences consisting of variable indices followed by variable values. Let $u$ be the uniform distribution over $\mathcal{S}$. Let $H(p, q)$ denote the cross entropy between distributions $p$ and $q$. We consider the following risk:*

$$R(q) = H(p, q) + H(u, q) \tag{4}$$

*Let $q^* = \arg\min_q R(q)$ be a minimizer of the risk over all possible probability distributions. Then, for all non-adjacent random variables $Y_i$ and $Y_j$, reasoning through the intermediate variables has lower bias than direct prediction. That is, for any $y_i, y_j \in \mathcal{X}$:*

$$|\mathbb{E}_{q_{\mathcal{S}}^*}[\hat{q}_S(Y_i = y_i|Y_j = y_j)] - p_d(Y_i = y_i|Y_j = y_j)|^2$$
$$< |\hat{q}_D(Y_i = y_i|Y_j = y_j) - p_d(Y_i = y_i|Y_j = y_j)|^2$$

Here, the expectation is over the randomness in the Monte Carlo samples of intermediate variables.

*Proof sketch.* We characterize $q^*$ in terms of $p_d$ and $u$ by analyzing the risk (Eq 4), which decomposes into a sum of cross-entropy terms, either between $q(\cdot)$ and $p(\cdot)$ or between $q(\cdot)$ and $u$.

We consider two cases. For any adjacent pairs $Y_i$ and $Y_{i+1}$, $q^*(Y_{i+1}|Y_i)$ will interpolate between the true conditional probability $p_d(Y_{i+1}|Y_i)$ and the uniform distribution. This is because the risk will contain a term of the form $-\mathbb{E}_{p_d(Y_{i+1}|Y_i)}[\log q(Y_{i+1}|Y_i)]$ and a term of the form $-\mathbb{E}_u[\log q(Y_{i+1}|Y_i)]$. The minimizer of the sum of these terms is a mixture between $p_d(Y_{i+1}|Y_i)$ and the uniform distribution. On the other hand, for non-adjacent pairs $Y_i$ and $Y_j$, $q^*(Y_i|Y_j) = \frac{1}{|\mathcal{X}|}$. As a consequence of the observation distribution $p_{\text{obs}}$, $Y_i$ and $Y_j$ will never appear together. There will only be a term of the form $-\mathbb{E}_u[\log q(Y_i|Y_j)]$ in the risk. Therefore, minimizing the risk for $Y_i$ and $Y_j$ is equivalent to minimizing the entropy regularization term since the risk does not penalize $q^*(Y_i|Y_j)$ for not matching $p_d(Y_i|Y_j)$.

Given $q^*$, we can calculate the bias of the scaffolded estimator $\hat{q}_S$, where we approximately marginalize out the intermediate random variables by sampling from the conditional probability distributions $q^*(Y_{i+1}|Y_i)$. Since these conditional distributions are mixture distributions, we can also show that the scaffolded estimator is a mixture between $p(Y_i|Y_j)$ and $u$ with mixture weight $\lambda \in (0, 1)$. Therefore, its bias can be expressed as $(1 - \lambda)^2 |p(Y_i|Y_j) - \frac{1}{|\mathcal{X}|}|$. The second term in the product is exactly the bias of the direct estimator and $(1 - \lambda)^2 < 1$. The desired inequality immediately follows. Intuitively, the bias of the scaffolded estimator is lower because it leverages the dependency structure of the data, chaining together local conditional probabilities of variables that appeared together in training to estimate conditional probabilities for unseen pairs. □

In practice, we find that density estimators tend to predict the marginal distributions for held-out pairs, *i.e.,* $q^*(X_i|X_j) \approx p(X_i)$. If we instead assume that the risk minimizer is a mixture between the true conditional distributions and the marginal distributions, we can relax some assumptions we make about the underlying data distribution. For further discussion, full details of the theorem, and proofs see Appendix A. In the next section, we explore whether the reasoning gap occurs in practice for transformer language models in situations with more complex dependency and observation structures.

## 4 Experimental methods

We test the hypothesis that locally structured training data leads to a reasoning gap by training a language model (i.e., an autoregressive density estimator with a transformer architecture) and testing conditional inference accuracy. We manipulate the structure of the training data and measure the effectiveness of different estimators. The key question is whether there are training conditions under

which scaffolded or free generation outperforms direct prediction. Code and data are available at `https://github.com/benpry/why-think-step-by-step`.

As in our theoretical analysis, we are interested in understanding the bias of different estimators. In practice, we evaluate our estimators using Mean Squared Error (MSE), averaging over 10 Monte Carlo samples for estimators that include intermediate variables. Averaging over samples reduces variance, meaning that the remaining estimation error is primarily driven by bias.

## 4.1 Training data

The first step in generating our training data is to create Bayes nets. We generate a random topology for the networks, creating 100 variables and incrementally adding 100 random edges between pairs of variables. After generating a topology, we create conditional probability tables for each variable. We design the conditional probability tables and select among generated Bayes nets to create non-adjacent pairs of variables with high mutual information. The probability of a variable being 1 for each setting of its parents is randomly sampled from $\text{Beta}(\frac{1}{5}, \frac{1}{5})$, which favors moderately strong dependencies. If we did not favor strong dependencies, conditional probabilities would be very close to marginal probabilities for non-adjacent pairs of variables and a language model could come very close to matching true conditional probabilities without learning the relationships between variables. Strong dependencies are therefore an important precondition to the effectiveness of reasoning. We generate 100 Bayes nets according to this process, then compute the mutual information between each pair of variables. We select the 50 pairs of non-adjacent variables that have the highest mutual information, then randomly sample 25 of those pairs to hold out from the training set. Finally, we select the 10 Bayes nets, out of the initial 100, for which the 25 held-out pairs of variables have the highest mean mutual information. Pseudocode for Bayes net generation is shown in Algorithm 1 of Appendix B.

For each of the 10 selected Bayes nets, we generate a training set consisting of 1 million samples formatted as strings. The strings consist of the name of each variable followed by its value; variables are ordered randomly. Variable names are the letter 'X' followed by a number from 0 to 99. For example, if the variable 'X42' has value 1, the string would include the line "X42=1". We mention the name of the last variable in the sample at the beginning of the string and mark it with the word "target." We also include '###' before the sample to indicate where it begins. A schematic example of a formatted sample string is shown in Figure 1B and a full example is shown in Appendix C.

Each sample includes only a subset of all the variables in the Bayes net, chosen according to the *observation distribution*, which is a distribution over subsets of variables. Pseudocode for selecting a subset of variables from an observation distribution is shown in Algorithm 2 in Appendix B. At a high level, observation distributions have three key properties:

**Locality**   Observed samples contain only variables from a local neighborhood, consisting of a central variable along with all variables of distance at most $k$ from it. To sample from the observation distribution, we sample the central variable uniformly randomly and then sample $k$ from some distribution that controls the local neighborhood size. In our experiments, we draw $k$ either from a geometric distribution with a parameter of $0.5$ or a Zipfian distribution with a parameter of $2$.

**Variable dropout**   Even within a local subset of the world, we may not see everything at once. Certain variables may be missing or go unnoticed. We formalize this intuition with *variable dropout*. With some probability ($0.2$ in our experiments), variables are dropped from a local neighborhood and the learner does not see them. Variable dropout may also help a model generalize to pairs of variables that were unseen in training as more subsets of variables appear in the training set.

**Held-out pairs**   Finally, some pairs of variables are held out across all training data. If a local neighborhood, after variable dropout, would include a pair of variables we decided to hold out, we randomly remove one of the two variables in the pair from the sample. We use the ability to match conditional probabilities for held-out pairs of variables, measured via mean squared error, as our main performance metric.

We also create two **control conditions** to compare against local observation distributions. As one control, we consider a training dataset in which the values of variables are sampled from one Bayes net, but the local neighborhoods are constructed with respect to the structure of a different Bayes net. This condition still has a local co-occurrence structure, but the co-occurrences do not reflect

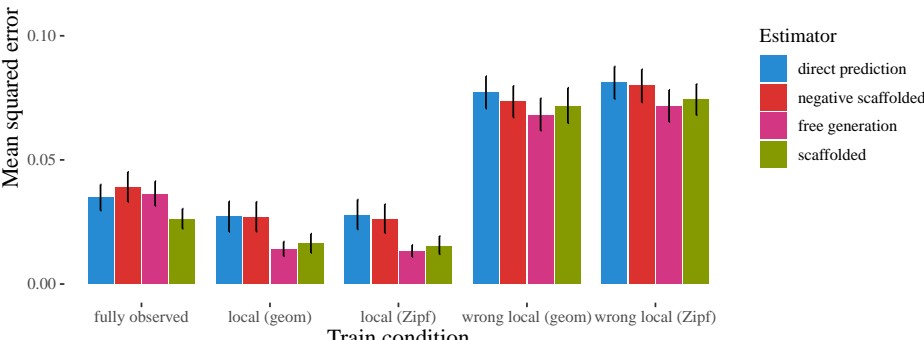

Figure 2: Mean squared error between estimated and true conditional probabilities for held-out pairs with high mutual information. Error bars denote 95% confidence intervals. Both free and scaffolded generation significantly outperform direct prediction when the training data is locally structured.

which variables influence each other. As another control, we use a *fully observed* condition where each sample contains almost all of the variables in the Bayes net. One of the two variables in each held-out pair is randomly dropped, but all other variables are included. These controls enable us to test whether local structure in the training data drives the value of reasoning.

## 4.2 Estimation

Each of the estimators in Section 2.2 is implemented via sampling from and scoring with the language model. For instance, in free generation, the model is prompted with the target variable name and initial observation, then sampled from until the target variable name is generated. The probabilities for possible values of the target variable are then examined. Figure 1C illustrates how we implement direct prediction and free generation as prompts and Appendix D contains examples of each estimator. For estimators that rely on Monte Carlo estimation, we use 10 samples.

We also introduce **negative scaffolded generation** as a control estimator that generates irrelevant intermediate variables. For each pair of variables, we select a random set of variables equal in size to the scaffold, but which does not include any of the scaffold variables. We prompt the language model to generate values for the negative scaffolds in the same way as in scaffolded generation.

## 4.3 Model architecture

We use a smaller version of the GPT-2 autoregressive transformer architecture [20], as implemented in the HuggingFace transformers library [21]. Our model has 512-dimensional embeddings, 10 layers, and 8 attention heads. Data is tokenized via a Byte Pair Encoding tokenizer fit to data in our format [22]. We trained this architecture with randomly initialized weights for $300,000$ gradient steps on batches containing $3,072$ tokens each, for a total of $921,600,000$ tokens of training. We trained models using the Adam optimizer [23]. Each model's training set consisted of $1,000,000$ samples from a single Bayes net. The models achieved near-perfect performance on the training task. Further training details are provided in Appendix E. We also compare multiple different architectures in Appendix F, demonstrating that our results are robust across them.

## 5 Results

Each held-out pair gives four data points: if we hold out the pair $X1, X2$ we can estimate $p(X1|X2 = 0), p(X1|X2 = 1), p(X2|X1 = 0)$, and $p(X2|X1 = 1)$. Pooling across the 10 Bayes nets we trained models on gives us $1,000$ data points per training condition and estimator. Figure 1D shows our main result: free generation performs better (lower MSE) than direct prediction and performs

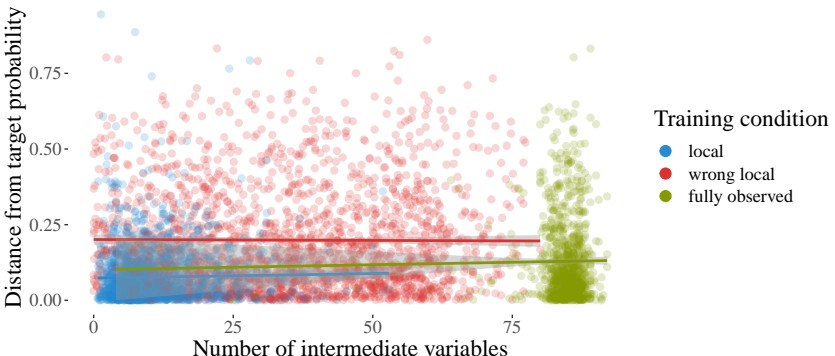

Figure 3: Number of intermediate variables generated in free generation vs. absolute distance between true and estimated conditional probabilities for each training condition. Lower dots indicate more accurate estimation, while dots further to the right indicate more intermediate variables generated.

Table 1: Mean squared errors with 95% confidence intervals between direct prediction probabilities and either true or marginal probabilities for non-held-out pairs with high mutual information. In the local training conditions, direct prediction is close to perfect at matching conditional probabilities.

|  | fully observed | local (geom) | local (Zipf) | wrong local (geom) | wrong local (Zipf) |
|---|---|---|---|---|---|
| true | .013 [.010, .016] | .004 [.003, .005] | .003 [.003, .004] | .052 [.045, .059] | .056 [.049, .063] |
| marginal | .080 [.073, .086] | .114 [.106, .121] | .112 [.104, .121] | .047 [.041, .052] | .033 [.029, .038] |

well with less training when the data is structured locally. The MSEs for all estimators and training conditions after 921.6 million tokens of training are shown in Figure 2.

Appendix G shows how MSE varies with the number of Monte Carlo samples taken for each estimator. Estimation error is high for free and scaffolded generation for low numbers of samples due to high variance, but quickly decreases with the number of samples. Re-sampling the intermediate variables is akin to self-consistency prompting methods which involve re-sampling the chain of thought and taking the majority or average answer [14].

## 5.1 When reasoning helps

We find a benefit to reasoning when the observation distribution has the correct locality structure. When the training data is structured locally with respect to strong dependencies, both scaffolded and free generation perform significantly better than direct prediction. We call this difference the *reasoning gap*. Scaffolded and free generation also perform significantly better than negative scaffolded generation, indicating that relevant intermediate variables help in predicting the target variable, but irrelevant intermediate variables do not. Furthermore, the success of free generation demonstrates that a model trained on local subsets of a Bayes net can self-generate helpful intermediate variables.

In free generation, the model generates variables until it produces the target variable. This results in reasoning traces of varying lengths, including many long traces. One might expect long traces to distract the model, but this did not occur in practice: the length of the reasoning trace does not relate strongly to accuracy within training conditions, as is shown in Figure 3. However, free generation does not outperform direct prediction substantially when the training data is fully observed.

It is also of note that training on locally structured data reduces the average length of reasoning traces (Fig. 3). In local training conditions, free generation usually produces a set of variables that $d$-separates the observed variable from the target variable – 70% of the time in the local neighborhood conditions. In contrast, the training conditions with the wrong locality structure only lead to $d$-separating reasoning traces 34% of the time. The fully observed training condition also usually creates $d$-separating reasoning traces (69%), likely because it generates a large number of variables. $d$-separation is important because it ensures that the intermediate variables capture all the influence that the observed variable exerts on the target variable. These results suggest that training on local clusters of variables is valuable in part because it makes it easy for an autoregressive model to

Table 2: Mean squared errors between estimated conditional probabilities and marginal probabilities of target variables with 95% confidence intervals. Lower values indicate closer matches.

| | fully observed | local (geom) | local (Zipf) | wrong local (geom) | wrong local (Zipf) |
|---|---|---|---|---|---|
| direct prediction | .086 [.079, .093] | .115 [.108, .124] | .116 [.107, .124] | .029 [.025, .033] | .022 [.019, .025] |
| negative scaffolded | .089 [.082, .096] | .116 [.108, .124] | .115 [.108, .123] | .036 [.031, .040] | .026 [.022, .030] |
| free generation | .123 [.115, .132] | .124 [.117, .132] | .127 [.120, .135] | .049 [.044, .055] | .042 [.037, .048] |
| scaffolded | .110 [.102, .117] | .134 [.126, .142] | .134 [.126, .142] | .051 [.046, .057] | .043 [.038, .049] |

naturally output useful intermediate reasoning steps. Still, chain-of-thought reasoning can be helpful even if the reasoning trace does not completely $d$-separate the two variables. In the local training conditions, the model does not generate a $d$-separating trace 30% of the time, yet there is virtually no performance difference between free and scaffolded generation.

## 5.2 Data complexity and reasoning

The combination of locally structured training data and step-by-step reasoning can also lead to accurate conditional probability estimates with much less training than would be required if a model were trained on fully observed data. To demonstrate this difference, we run both direct prediction and free generation on checkpoints from our models that are saved after every 5000 gradient steps (15.3 million tokens) of training. Part D of Figure 1 shows the results of this analysis.

When data is fully observed, direct prediction and free generation performances improve slowly but consistently. There is no reasoning gap, as both estimators perform almost identically. When the model is trained on data from geometrically sized local neighborhoods, direct prediction performance improves quickly, then plateaus. There is a substantial reasoning gap where free generation outperforms direct prediction. Free generation matches the true conditional probabilities very closely after about 120 million training tokens, achieving the best performance of any combination of training condition and estimator. Still, a sufficiently large transformer can eventually memorize the relationships in its training corpus with enough training. We evaluate the data complexity of direct learning by training a language model of the same architecture on fully observed data without holding out any pairs. Direct prediction in this case takes over 3 times as much training to achieve the same performance as local training with free generation, despite the model seeing the relevant pairs in training (see Appendix H for detailed results). Training language models on datasets consisting of local neighborhoods with strong dependencies and performing chain-of-thought reasoning at inference time can therefore be more data-efficient than training on more complete datasets.

## 5.3 When reasoning is unnecessary

Table 1 shows MSEs between direct prediction estimates and true probabilities for the 25 high-mutual-information variable pairs that were *not* held out from the training set. In the local training conditions (and to a lesser extent fully observed), direct prediction matches the true conditional probabilities almost perfectly. This result indicates that our language models learn to match conditional probabilities directly when the observed and target variables co-occur in the training distribution. Observing these variables together frequently obviates the need for step-by-step reasoning.

## 5.4 When reasoning fails

We test the hypothesis that language models match the *marginal* probabilities of the target variables when they fail to predict the true conditional probabilities by comparing models' estimated conditional probabilities against the true marginal probabilities. The MSEs between the estimated conditional probabilities and marginal probabilities of the relevant target variables are shown in Table 2. We can see the opposite of the trend for true probabilities: the worse a training condition does at matching the true conditional probability, the better it matches the marginal. The language models trained on data with the wrong locality structure generated estimates that were particularly close to the marginal probabilities. When the variables that co-occur with each other frequently are not local in the Bayes net, they often have very little influence on each other. This means that the joint distribution over co-occurring variables is usually very close to the product of the marginal probabilities, i.e.

$P(X_1, X_2, X_3) \approx P(X_1)P(X_2)P(X_3)$ for non-local $X_1, X_2, X_3$. Without the ability to estimate conditional probabilities accurately, there are no reliable 'steps' for step-by-step reasoning to use.

## 6 Discussion

Our theoretical and experimental results demonstrate that estimating conditional probabilities by reasoning through intermediate variables can outperform direct prediction when the training data has local structure. When the training data includes all the variables, free generation does not lead to better performance than direct prediction. When the training data has the wrong locality structure, the models do not reliably learn conditional probabilities that can be chained together.

Our results provide a minimal case in which chain-of-thought reasoning is helpful and suggest conditions under which it is likely to be helpful in more naturalistic settings: we can expect chain-of-thought reasoning to help when a model is tasked with making inferences that span different topics or concepts that do not co-occur often in its training data, but can be connected through topics or concepts that do. Re-sampling the chain of thought multiple times might be especially valuable in domains with moderately strong dependencies. In these settings, introducing intermediate steps increases the variance of estimation and re-sampling multiple times reduces it. We have also shown that chain-of-thought reasoning enables greater data efficiency: a model trained on locally structured data with chain-of-thought reasoning at inference time can accurately match conditional probabilities with less training than a model trained on all the data. This result can provide insight into why humans are more data-efficient than language models. Since humans experience the world from a first-person perspective, the information we encounter is structured in small clusters of entities that are tightly coupled with each other. This idea could also be relevant to data curation for the training of language models. Constructing datasets with tightly correlated observation neighborhoods that collectively cover the space of relevant concepts may amplify language models' ability to perform chain-of-thought reasoning while reducing their data needs.

Future work should explore the structure of the observation distribution for human learners to understand how the information we observe facilitates reasoning. Using modern language models as toy models for the study of reasoning is a promising direction for cognitive science to address long-standing problems in reasoning and problem-solving, such as the value of thought experiments.

The role of abstraction in reasoning is another important direction for future work. Humans often use abstract principles to reason rather than exclusively considering possible concrete states of the world (settings of observable variables). Likewise, the language corpora that language models are trained on contain abstract concepts expressed in natural language. Understanding how abstractions can arise and the effects they have on reasoning would be helpful in connecting our setup to more naturalistic reasoning contexts. The possibility of taking abstract steps makes the problem of choosing useful reasoning steps much harder, so we plan to explore questions of how language models might learn to reason more effectively through fine-tuning, in-context learning, and reinforcement learning.

**Limitations** Of the many forms of chain-of-thought prompting that exist in the literature, our findings are most relevant to zero-shot prompting [e.g. 9] where a model produces steps on the way to an answer without any other examples of reasoning traces in its context window. It is less applicable to methods that involve giving the model examples of reasoning [e.g. 10]. Our results are also in the context of simple propositional worlds. Reasoning in rich, structured worlds may require more expressive languages with which to reason and specify hypotheticals. Our theoretical results do not explain the phenomenon observed in practice that transformers revert to the marginal distributions for held-out pairs. We leave it to future work to explore how the inductive biases of transformers might induce this behavior on unseen data.

## Acknowledgments and Disclosure of Funding

We thank members of the Computation and Cogniton Lab, the Causality in Cognition Lab, and the Stanford NLP Group for feedback on earlier versions of this work. In particular, we thank Tobias Gerstenberg for advice on the importance of strong dependencies for reasoning and Eric Zelikman for feedback on the design of our figures. This work is supported by an NSF Expeditions Grant, Award Number (FAIN) 1918771.

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

# A  Theoretical analysis

In this section, we theoretically analyze chain-of-thought reasoning when the underlying data distribution can be represented as a directed chain. We prove that any minimizer of a risk consisting of a cross-entropy term and an entropy regularization term will exhibit a "reasoning gap" when the data has a particular locality structure.

## A.1  Problem Setup

We assume that we observe a sequence of *variable indices* $i_t \in \{1, \ldots, N\}$ and *variable values* $v_t \in \mathcal{X}$ for some finite set $\mathcal{X}$. In the sequence, variable values always follow variable indices. Since our results concern estimators that are only ever conditioned on a single variable, it suffices to consider sequences of length two. However, our setup can be interpreted as assuming we observe independent local neighborhoods of size 2 with token separators.

**Factorization of data distribution**  We assume that the joint distribution $p_d$ over $Y_1, Y_2, \ldots, Y_N$ factorizes as

$$p_d(Y_1, \ldots Y_N) = p_d(Y_1) \prod_{j=1}^{N} P_d(Y_{j+1}|Y_j)$$

and that $Y_1, Y_2, \ldots, Y_N$ take support in some finite set $\mathcal{X}$. Note that the variable indices index particular random variables in $p_d$.

**Local observation distribution**  We will consider a setting where only adjacent variables in the graphical model can appear adjacent in the sequence. More formally, let $\mathcal{Y}$ refer to the set of all variable identifiers. Let $\mathcal{Y}_{\text{obs}}$ be the set of all allowed pairs of variable indices. Then, $\mathcal{Y}_{\text{obs}} \subseteq \mathcal{P}(\{1, \ldots, N\})$ and $\mathcal{Y}_{\text{obs}}$ consists of pairs of the form $(i, i+1)$. We will define the distribution over these valid subsets of variable indices as $p_{\text{obs}}$; in particular, $p_{\text{obs}}$ takes support only on $\mathcal{Y}_{\text{obs}}$ and assigns equal probability to all of its pairs. The two distributions $p_d$ and $p_{\text{obs}}$ define a distribution $p$ over the sequence of variable indices and variable values.

That is, for any non-adjacent pairs $Y_i$ and $Y_j$ and for any variable values $y_i$ and $y_j$,
$$p(i_1 = i, v_1 = y_i, i_2 = j, v_1 = y_j) = 0 \tag{5}$$
For adjacent pairs $Y_{i+1}$ and $Y_i$ and for variable values $y_{i+1}$ and $y_i$,
$$p(i_1 = i, v_1 = y_i, i_2 = i+1, v_1 = y_{i+1}) \propto p_d(Y_i = y_i, Y_j = y_j) \tag{6}$$

Intuitively, this property will imply that the risk minimizer only gets to see a small subset of local statistical dependencies in the graphical model. This key property of the observation distribution will induce a reasoning gap.

## A.2  Preliminaries

We begin with a proposition that will be useful in the main theorem. The proposition concerns the minimizer of the sum of two cross-entropy terms.

**Proposition 1.** *Let* $\alpha_1 \geq 0, \alpha_2 \geq 0$. *Let* $R(q) = \alpha_1 \mathbb{E}_{p_1(x)}[-\log q(x)] + \alpha_2 \mathbb{E}_{p_2(x)}[-\log q(x)]$. *Then* $q^* = \arg\min R(q) = \frac{\alpha_1}{\alpha_1+\alpha_2} p_1(x) + \frac{\alpha_2}{\alpha_1+\alpha_2} p_2(x)$

*Proof.* We assume the probability distributions are discrete. The Lagrangian is given by
$$\mathcal{L}(q, \lambda_0) = -\alpha_1 \sum_x p_1(x) \log q(x) + -\alpha_2 \sum_x p_2(x) \log q(x) + \lambda_0 \left(\sum_x q(x) - 1\right) \tag{7}$$

The first-order conditions are
$$\frac{\partial \mathcal{L}}{\partial q(x)} = -\alpha_1 \frac{p_1(x)}{q(x)} + -\alpha_2 \frac{p_2(x)}{q(x)} + \lambda_0 = 0 \tag{8}$$
$$\frac{\partial \mathcal{L}}{\partial \lambda_0} = \sum_x q(x) - 1 = 0 \tag{9}$$
$$\tag{10}$$

The first condition allows us to write $q(x) = \frac{\alpha_1 p_1(x) + \alpha_2 p_2(x)}{\lambda_0}$. We can substitute this for $q$ into the other first order condition (*i.e., q* normalizes) to obtain

$$\sum_x \frac{\alpha_1 p_1(x) + \alpha_2 p_2(x)}{\lambda_0} = 1$$

This implies $\lambda_0 = \alpha_1 + \alpha_2$ and the desired result follows immediately. $\qquad \square$

### A.3  Main Theorem

Our main result is that the minimizer of the risk $q^*$ under the distribution $p$ defined by $p_{\text{obs}}$ and $p_d$ and a cross-entropy loss with entropy regularization will exhibit a "reasoning gap". We begin with a characterization of $q^*$. For simplicity, in the result below, we'll use $x_t$ to denote elements of the sequence that correspond either to variable values $v_t$ or variable indices $i_t$.

**Theorem A.1.** *Let $u$ be the uniform distribution. Let $p$ be the distribution over variable indices and variable values defined by $p_{obs}$ and $p_d$. Let $H(p, q)$ denote the cross entropy between distributions $p$ and $q$. We consider the following risk:*

$$R(q) = H(p, q) + H(u, q) \tag{11}$$

*Then $q^* = \arg\min_q R(q)$ satisfies the following properties.*

- *For all pairs of adjacent random variables $Y_i$ and $Y_{i+1}$, $q^*(Y_i|Y_j) = \lambda p_d(Y_{i+1}|Y_i) + (1 - \lambda)\frac{1}{|\mathcal{X}|}$ for some $\lambda \in (0, 1)$.*

- *For all pairs of non-adjacent random variables, $q^*(Y_i|Y_j) = \frac{1}{|\mathcal{X}|}$.*

*Proof.* We can write the risk as a sum across timesteps.

$$R(q) = \sum_{t=1}^T \mathbb{E}_{p(x_{1:t})}[-\log q(x_t|x_{1:t-1})] + \sum_{t=1}^T \mathbb{E}_{u(x_{1:t})}[-\log q(x_t|x_{1:t-1})] \tag{12}$$

$$\tag{13}$$

Let us consider the left sum. By the law of iterated expectations, we can decompose the fourth term in the sum where $x_4$ is a variable value as

$$\mathbb{E}_{p(i_{1:2}, v_{1:2})}[-\log q(v_2|i_{1:2}, v_1)] \tag{14}$$

$$= \mathbb{E}_{p(i_{1:2}, v_1)}[\mathbb{E}_{p(v_2|i_{1:2}, v_1)}[-\log q(v_2|i_{1:2}, v_1)]] \tag{15}$$

$$= \sum_{i_1}\sum_{v_1}\sum_{i_2} \mathbb{E}_{p(v_2|i_{1:2}, v_1)}[-\log q(v_2|i_{1:2}, v_1)]p(i_{1:2}, v_1) \tag{16}$$

$$= \sum_{(i_1, i_2) \in \mathcal{Y}_{\text{obs}}}\sum_{v_1} \mathbb{E}_{p(v_2|i_{1:2}, v_1)}[-\log q(v_2|i_{1:2}, v_1)]p(i_{1:2}, v_1) \tag{17}$$

Importantly, notice that the outer sum over the variable indices $i_1, i_2$ is over $\mathcal{Y}_{\text{obs}}$ due to the observation distribution.

On the other hand, consider the right sum.

$$\mathbb{E}_{u(i_{1:2}, v_{1:2})}[-\log q(v_2|i_{1:2}, v_1)] \tag{18}$$

$$= \mathbb{E}_{u(i_{1:2}, v_1)}[\mathbb{E}_{u(v_2|k_{1:2}, v_1)}[-\log q(v_2|i_{1:2}, v_1)]] \tag{19}$$

$$= \sum_{i_1}\sum_{v_1}\sum_{i_2} \mathbb{E}_{u(v_2|i_{1:2}, v_1)}[-\log q(v_2|i_{1:2}, v_1)]u(i_{1:2}, v_1) \tag{20}$$

$$= \sum_{(i_1, i_2) \in \mathcal{Y} \times \mathcal{Y}}\sum_{v_1} \mathbb{E}_{u(v_2|i_{1:2}, v_1)}[-\log q(v_2|i_{1:2}, v_1)]u(i_{1:2}, v_1) \tag{21}$$

$$\tag{22}$$

We consider two different cases. Suppose the variable indices are adjacent so that $i_2 = i + 1$ and $i_1 = i$ for some $i$. In addition, fix some value for $v_1$. Then, we can take exactly two terms, one from the left and right sum.

$$\frac{1}{2}\mathbb{E}_{p(v_2|i_{1:2},v_1)}[-\log q(v_2|i_{1:2},v_1)]p(i_{1:2},v_1)$$
$$+\frac{1}{2}\mathbb{E}_{u(v_2|i_{1:2},v_1)}[-\log q(v_2|i_{1:2},v_1]u(i_{1:2},v_1) \tag{23}$$

The expectations that appear in Eq 23 are either 1) a cross entropy between either $q(Y_i|Y_j)$ and $p_d(Y_i|Y_j)$ or 2) a cross entropy between $q(Y_i|Y_j)$ and the uniform distribution. By an application of Proposition 1, the sum of these two terms is minimized by taking $q^*(Y_i|Y_j) = \lambda_{i,j}\frac{1}{|\mathcal{X}|} + (1 - \lambda_{i,j})p(Y_i|Y_j)$ for $\lambda_{i,j} = \frac{u(i_{1:2},v_1)}{u(i_{1:2},v_1)+p_d(Y_i|Y_j)}$. The result holds for any adjacent pairs.

Finally, we consider non-adjacent pairs $Y_i$ and $Y_j$. By construction, we can never have $i_2 = i, i_1 = j$ in the left sum. We will only have one cross entropy term between $q^*(Y_i|Y_j)$ and $u(Y_i)$. Therefore, minimizing the risk is equivalent to minimizing the entropy regularization term and taking $q^*(Y_i|Y_j) = \frac{1}{|\mathcal{X}|}$. $\qquad\square$

In the following corollary, we will show that the squared bias of the scaffolded generation estimator, which explicitly marginalizes out intermediate variables, is less than the squared bias of the direct estimator. For this result, we assume that the true conditional probabilities satisfy a doubly stochastic condition; that is, $\sum_{y_i} p(Y_i = y_i|Y_j = y_j) = 1 = \sum_{y_j} p(Y_i = y_i|Y_j = y_j)$. We discuss how to relax this assumption under different assumptions about the risk minimizer.

**Theorem A.2.** *For all $y_i, y_j \in \mathcal{X}$ with $|i - j| > 1$,*

$$|\mathbb{E}[\hat{q}_S^*(Y_i = y_i|Y_j = y_j)] - p(Y_i = y_i|Y_j = y_j)|^2 < |\hat{q}_D^*(Y_i = y_i|Y_j = y_j) - p(Y_i = y_i|Y_j = y_j)|^2$$

*Proof.* First, we show that $\mathbb{E}[\hat{q}_S^*(Y_i = y_i|Y_j = y_j)] = \lambda p(Y_i = y_i|Y_j = y_j) + (1 - \lambda)\frac{1}{|\mathcal{X}|}$ for some $\lambda \in (0, 1)$. Here the expectation is taken with respect to $y_i \sim q^*(Y_i|Y_{i-1} = y_{i-1}), y_{i-1} \sim q^*(Y_{i-1}|Y_{i-2} = y_{i-2})\ldots y_{j+1} \sim q^*(Y_{j+1}|Y_j = y_j)$. For conciseness, in our notation, we omit the random variables that the expectation is taken with respect to.

We prove this by induction on $|i - j|$, the distance between the nodes corresponding to $Y_i$ and $Y_j$ in the directed chain. We consider the base case $|i - j| = 2$.

$$\mathbb{E}[\hat{q}_S^*(Y_3 = y_3|Y_1 = y_1)] = \mathbb{E}[(1 - \lambda_{3,2})p(Y_3 = y_3|Y_2 = y_2) + \lambda_{3,2}\frac{1}{|\mathcal{X}|}]$$

$$= \lambda_{3,2}\frac{1}{|\mathcal{X}|} + (1 - \lambda_{3,2})\mathbb{E}[p(Y_3 = y_3|Y_2 = Y_2)]$$

$$= \lambda_{3,2}\frac{1}{|\mathcal{X}|} + (1 - \lambda_{3,2})\sum_{y_2} p(Y_3 = y_3|Y_2 = y_2)[(1 - \lambda_{2,1})p(Y_2 = y_2|Y_1 = y_1) + \lambda_{2,1}\frac{1}{|\mathcal{X}|}]$$

$$= \lambda_{3,2}\frac{1}{|\mathcal{X}|} + (1 - \lambda_{3,2})(1 - \lambda_{2,1})\sum_{y_2} p(Y_3 = y_3|Y_2 = y_2)p(Y_2 = y_2|Y_1 = y_1)$$

$$+ (1 - \lambda_{3,2})\lambda_{2,1}\sum_{y_2}\frac{1}{|\mathcal{X}|}p(Y_3 = y_3|Y_2 = y_2)$$

$$= \lambda_{3,2}\frac{1}{|\mathcal{X}|} + (1 - \lambda_{3,2})(1 - \lambda_{2,1})p(Y_3 = y_3|Y_1 = y_1)$$

$$+ (1 - \lambda_{3,2})\lambda_{2,1}\frac{1}{|\mathcal{X}|}$$

$$= (1 - \lambda)\frac{1}{|\mathcal{X}|} + \lambda p(Y_3 = y_3|Y_1 = y_1)$$

Here $\lambda = (1 - \lambda_{3,2})(1 - \lambda_{2,1})$. Importantly, the expected value is a strict convex combination of the uniform and conditional probabilities. For the induction step, we note that the expectations $\mathbb{E}[\hat{q}_S^*(Y_i = y_i | Y_j = y_j)]$ and $\mathbb{E}[\hat{q}_S^*(Y_{i-1} = y_{i-1} | Y_j = y_j)]$ are related as follows

$$\mathbb{E}[\hat{q}_S^*(Y_i = y_i | Y_j = y_j)] = \lambda \frac{1}{|\mathcal{X}|} + (1 - \lambda) \sum_{y_{i-1}} p(Y_i = y_i | Y_{i-1} = y_{i-1}) \times \mathbb{E}[\hat{q}_S^*(Y_{i-1} = y_{i-1} | Y_j = y_j)]$$

We now use this characterization of the scaffolded estimator as a mixture distribution to prove the desired inequality. The bias of direct estimator can be computed as $|\frac{1}{|\mathcal{X}|} - p(Y_i = y_i | Y_j = y_j)|^2$. By the previous results, $\mathbb{E}[\hat{q}_S^*(Y_i = y_j | Y_i = y_i)] = \lambda p(Y_i | Y_j) + (1 - \lambda) \frac{1}{|\mathcal{X}|}$ for some $\lambda \in (0, 1)$.

$$|\mathbb{E}[\hat{q}_S^*(Y_i | Y_j = y_j)] - p(Y_i = y_i | Y_j = y_j)|^2$$

$$= |\lambda p(Y_i = y_i | Y_j = y_j) + (1 - \lambda) \frac{1}{|\mathcal{X}|} - p(Y_i = y_i | Y_j = y_j)|^2$$

$$= |\lambda p(Y_i = y_i | Y_j = y_j) + \frac{1}{|\mathcal{X}|} - \lambda \frac{1}{|\mathcal{X}|} - p(Y_i = y_i | Y_j = y_j)|^2$$

$$= |(\lambda - 1) p(Y_i = y_i | Y_j = y_j) + (1 - \lambda) \frac{1}{|\mathcal{X}|}|^2$$

$$= (1 - \lambda)^2 |(\frac{1}{|\mathcal{X}|} - p(Y_i = y_i | Y_j = y_j)|^2$$

$$< |\frac{1}{|\mathcal{X}|} - p(Y_i = y_i | Y_j = y_j)|^2$$

Interestingly, if we assumed that the risk minimizer interpolates between the marginals and the true conditionals, we could relax the doubly stochastic condition.

$\square$

## B    Pseudocode for data generation

This section contains pseudocode for the algorithms we use to generate Bayes nets and select variables to include in a given sample according to the observation distribution.

First, we use Algorithm 1 to create Bayes nets. This algorithm takes in a number of nodes and edges to create and defines a directed acyclic graph by randomly adding edges between pairs of nodes. Next, it assigns conditional probability tables to the nodes given the values of their parents to create a Bayes net.

For each sample, we only show a subset of all the variables according to an observation distribution. Algorithm 2 shows the procedure we use to select which variables to display in a given sample. We first sample central variable $c$ (uniformly) and a distance $k$ (according to a geometric or Zipfian distribution), then get all the variables within distance $k$ of $c$ in the graph. Next, we remove each variable with probability 0.2. If any of the held-out pairs remain, we randomly remove one variable from each pair. In the wrong locality structure training conditions, we use a graph $G$ that does not correspond to the net from which our samples are drawn but has the same variable names. In the fully observed condition, we start with all variables, but still remove held-out pairs using the final for loop.

## C    Full sample of training data

The following is an example of a sample from a local neighborhood of a Bayes net in string format, as used in training our models:

```
###
target: X5
X17=0
X92=0
X13=0
```

**Algorithm 1** Algorithm for generating a Bayes net

**Input:** Number of nodes $N$, number of edges $M$
$G = (E, V) \leftarrow$ empty graph
**for** $i \in \{1, \ldots, N\}$ **do**
   $V \leftarrow V \cup \{Xi\}$
**end for**
**for** $i \in \{1, \ldots, M\}$ **do**
   $v_1, v_2 \leftarrow$ random pair of vertices $\in V$
   **while** $(v_1, v_2) \in E$ or $(v_1, v_2)$ would create a cycle **do**
      $v_1, v_2 \leftarrow$ random pair of vertices $\in V$
   **end while**
   $E \leftarrow E \cup \{(v_1, v_2)\}$
**end for**
$T \leftarrow$ empty associative array
**for** $v \in \text{TOPOLOGICALSORT}(G)$ **do**
   $t \leftarrow$ empty conditional probability table
   **for** $c \in$ possible configurations of $v$'s parents **do**
      $p \leftarrow \text{SAMPLE}(\text{BETA}(\frac{1}{5}, \frac{1}{5}))$
      $t[c] \leftarrow p$
   **end for**
   $T[v] \leftarrow t$
**end for**
$D \leftarrow$ Bayesian network defined by graph $G$ and conditional probability tables $T$
**return** $D$

**Algorithm 2** Algorithm for selecting variables to include based on an observation distribution

**Input:** Bayes net graph $G = (V, E)$, set of held-out pairs $P$, size distribution $D$
$c \leftarrow$ random variable $\in V$
$k \leftarrow \text{SAMPLE}(D)$
$R \leftarrow \{\}$
**for** $v \in V$ **do**
   **if** $v$ is within distance $k$ of $c$ in $G$ **then**
      $R \leftarrow R \cup \{v\}$
   **end if**
**end for**
**for** $v \in R$ **do**
   **if** $\text{SAMPLE}(\text{Bernoulli}, 0.2) = 1$ **then**
      $R \leftarrow R \setminus \{v\}$
   **end if**
**end for**
**for** $(v_1, v_2) \in P$ **do**
   **if** $v_1 \in R$ and $v_2 \in R$ **then**
      **if** $\text{SAMPLE}(\text{Bernoulli}, 0.5) = 1$ **then**
         $R \leftarrow R \setminus \{v_1\}$
      **else**
         $R \leftarrow R \setminus \{v_2\}$
      **end if**
   **end if**
**end for**
**return** $R$

```
X52=1
X24=1
X26=1
X91=0
X36=0
X34=0
X12=1
X20=0
X5=1
```

The full training set consists of $1,000,000$ samples like this concatenated together.

## D   Formatting estimators as prompts

### D.1   Direct prediction

In direct prediction, we use a simple prompt that specifies the label and value for the observed variable then states the name of the target variable. For example, if we wanted to estimate $p(X_2|X_1 = 0)$ we would use the following prompt:

```
###
target: X2
X1=0
X2=
```

We would then take the softmax of the log probabilities the language model assigns to '1' and '0' being the next token and use the resulting probability of '1' as the estimate.

### D.2   Scaffolded generation

In scaffolded generation, we pre-compute a sequence of intermediate variables, then use the language model to estimate their values. For example, if we wanted to estimate $p(X_4|X_1)$ and we knew $X_2$ and $X_3$ were scaffold variables, we would start with the following prompt:

```
###
target: X4
X1=0
X2=
```

We would then sample the next token (either $0$ or $1$) from the language model and append it to the prompt, followed by the name of the next scaffold variable on a new line. For example, if we got a value of $1$ from the language model, we would give it the following prompt next:

```
###
target: X4
X1=0
X2=1
X3=
```

We repeat this process until all scaffold variables have values, then compute the probability assigned to the target variable at the end, in the same way we do for direct prediction. The final prompt we use to estimate the probability might look like this:

```
###
target: X4
X1=0
X2=1
X3=0
X4=
```

We repeat this sampling process 10 times to produce a Monte Carlo estimate over the values of the scaffold variables and average the probabilities assigned to the target variable over samples.

### D.3 Free generation

Free generation is similar to scaffolded generation, but we sample from the model to choose *which* intermediate variables to instantiate, rather than just their values. For example, if we were inferring $p(X_4|X_1 = 0)$ we would first prompt the model like this:

```
###
target: X4
X1=0
```

We then sample the next two tokens from the model, which is exactly enough for one variable name. We add an equals sign, then sample the variable's value in the same way we do in other estimators. For example, our prompt might look like this after generating one intermediate variable and its value

```
###
target: X4
X1=0
X5=1
```

We repeat this process until the model outputs the name of the target variable. At that point, we have a prompt that might look like this

```
###
target: X4
X1=0
X5=1
X2=0
X7=0
X3=1
X4=
```

We then extract the probability of the target variable as in other estimators. Again, we average target variable probabilities over 10 samples of the intermediate variables.

## E  Training details

The main architecture we used is a smaller version of the GPT-2 architecture, with 512-dimensional embeddings, 10 layers, and 8 attention heads. It has $32,573,440$ parameters in total. We fit a Byte Pair Encoding tokenizer to data from our samples, which produced 356 unique tokens.

Our text was grouped into chunks of 1024 tokens and batches of 3, for a total of 3072 tokens per gradient step. We used the Adam optimizer, with an initial learning rate of $10^{-3}$ and Beta values of 0.9 and 0.999.

All models were trained on Nvidia Titan Xp GPUs. They were trained for $300,000$ gradient steps, which took approximately 20 hours each.

## F  Comparison of reasoning gaps across different architectures

To ensure that our results are robust across different transformer architectures, we compare mean squared error by estimator for four alternative architectures. We focus on training data consisting of geometrically-sized local neighborhoods. Figure 4 shows mean squared error by architecture type trained on geometrically-sized local neighborhoods of variables. Results are very similar across all architectures, except for the tiny architecture which fails to match the distribution at all.

We compare the original architecture to a smaller ($4,473,600$-parameter) architecture, a larger ($86,628,864$-parameter) architecture, and a wider architecture with larger embeddings but fewer

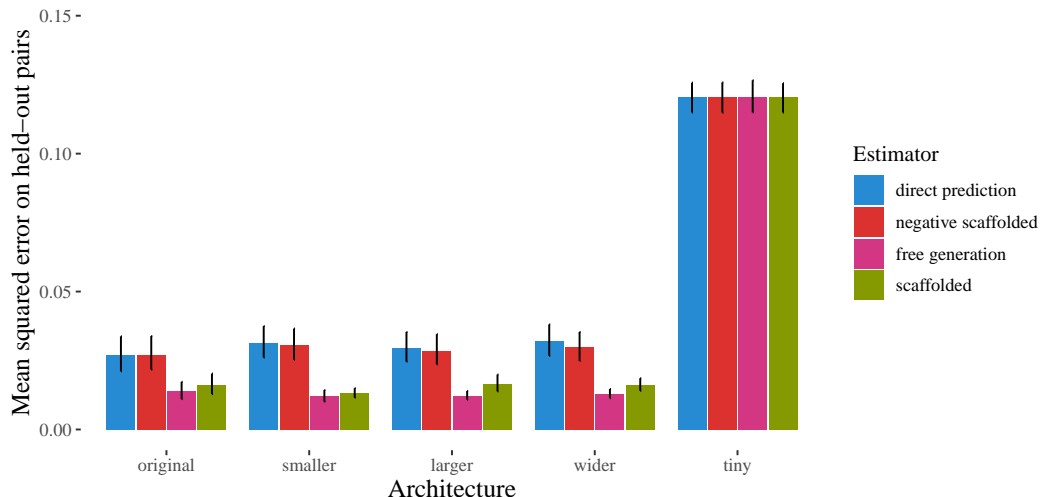

Figure 4: Mean squared error by architecture type and estimator. All architectures were trained on the same data consisting of geometrically-sized local neighborhoods of data. Error bars indicate 95% confidence intervals.

Table 3: Embedding size, number of layers, and number of attention heads for each of the five architectures used in the architecture comparison

| Architecture type | Embedding Size | # Layers | # Attention Heads |
|:---:|:---:|:---:|:---:|
| original | 512 | 10 | 8 |
| smaller | 256 | 5 | 4 |
| larger | 768 | 12 | 12 |
| wider | 1024 | 3 | 8 |
| tiny | 4 | 2 | 2 |

layers ($39,887,872$ parameters). Finally, we compare against a tiny architecture that is too small to match the distribution of the true Bayes net ($8,644$ parameters). Table 3 shows the embedding sizes, numbers of layers, and numbers of attention heads for each architecture.

## G  Mean squared error by number of samples

As mentioned in the main text, the number of Monte Carlo samples taken over intermediate variables influences the variance of the scaffolded and free generation estimators. We demonstrate this computationally by reporting the results of each estimator for different numbers of samples for the model trained on geometrically-sized local neighborhoods of variables. Figure 5 shows MSE by number of samples using each estimator that averages over samples, compared against the baseline of direct prediction. Free and scaffolded generation both have higher MSE than direct prediction with one or two samples due to the increase in variance. However, they achieve lower MSE when averaging over more than three samples. Negative scaffolded generation exhibits very similar performance to direct prediction for all numbers of samples, which suggests that the values of intermediate variables do not influence the final prediction for that estimator.

## H  Data efficiency of fully observed training with no held-out pairs

To supplement our data efficiency analyses, we ran a version of the fully observed training condition without any held-out pairs. In this setting, the transformer can directly memorize the true conditional probabilities, but it takes a long time for direct prediction to perform as well as free generation with locally-structured data. Figure 6 compares the performance of direct prediction and free generation

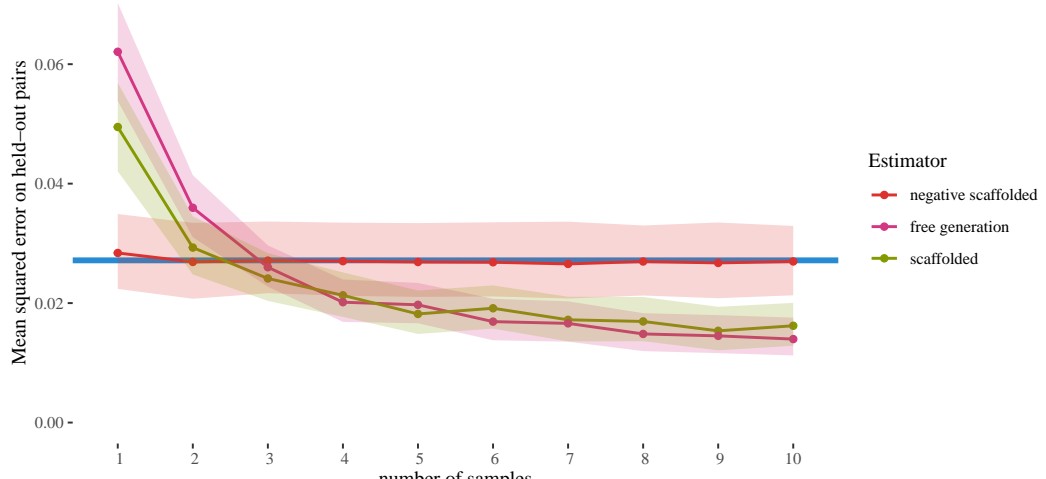

Figure 5: Mean squared error by number of Monte Carlo samples of intermediate variables for models trained on geometrically-sized local neighborhoods of variables. The blue line indicates the MSE achieved by direct prediction. Ribbons indicate 95% confidence intervals.

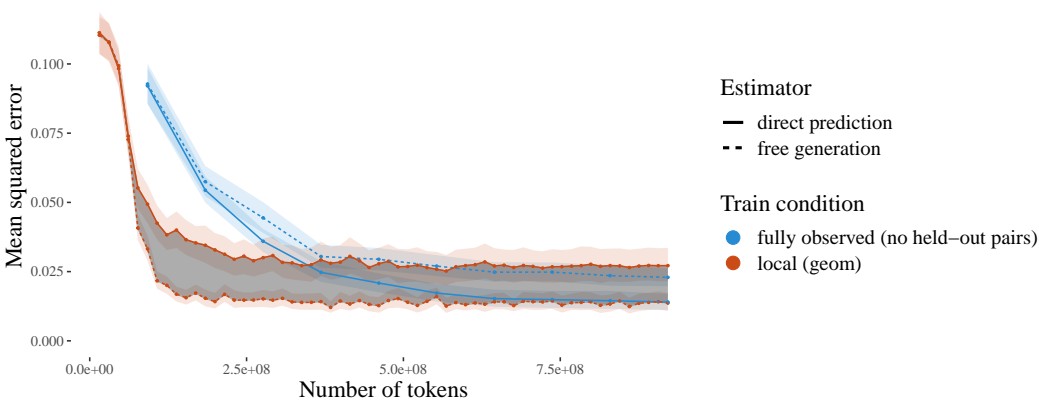

Figure 6: Learning curves comparing mean squared error on held-out pairs, estimated using free and direct prediction for training data consisting of either geometrically-sized local neighborhoods or the full set of variables. Unlike the version reported in the main text, no pairs of variables are held out in this fully observed condition. Even though the model is trained directly on the held-out pairs in the fully observed condition, there is a substantial data efficiency advantage to using locally structured training data and free generation at inference time.

for locally structured training data with held-out pairs and fully observed training data with no held-out pairs. Direct prediction with fully observed training data can outperform free generation with locally-structured training data. However, it takes approximately 650 million tokens of training to reach the same performance that free generation with geometric locally-structured training data converges to after only about 200 million tokens of training.

