# OpenReview forum: "Why think step by step? Reasoning emerges from the locality of experience"
_NeurIPS.cc/2023/Conference — NeurIPS 2023 oral_

### Official Review · Reviewer_jyTS · 2023-07-06

**Soundness:** 3 good
**Presentation:** 3 good
**Contribution:** 3 good
**Rating:** 8
**Confidence:** 3

**Summary:**

This paper aims to investigate in a control and toy setup why it is that zero-shot chain-of-thought reasoning (e.g. prompting a model with "let's think step by step" and letting the model output intermediate reasoning traces before generating the final answer) improves downstream performance of language models in reasoning tasks. The authors hypothesise that reasoning is useful (or even necessary) when there is a local structure in the training data. A local structure in the training data here refers to local clusters of variables that directly influence each other in a DAG are observed together. The high-level mapping of this onto real-world experience is that we perceive things that are physically and temporally close, but nonetheless can reason over things that are physically and temporally far away.

The setup in which the authors investigate this is a Bayes net. Imagine a directed acyclic graph (DAG) of variables, and the task is the predict the probability that one variable (the target) takes a value given that another variable (the input; physically separated from the target variable in the graph) has a certain value. The training setup is such that the learner does not see the target and the input variable together during training, but sees local clusters of variables from the graph. Importantly, these clusters overlap. So for each cluster there is at least one variable that is also in another cluster. The authors then show that a language model that is allowed to freely generate intermediate variables can better predict the target value probability than a model that directly predicts it, but only if the training data is structured in the local way described above. Intuitively, the learner achieves this by generating one of the overlapping variables and then moving between local clusters it has seen from training until it encounters the target variable. The authors also show that the learner's performance does not deteriorate with length of the intermediate variables generated, and that a learner that reasons freely is more data-efficient.

The authors take these results to conclude that we can expect zero-shot CoT reasoning to help when a learner has to make inferences over things (concepts, topics, you name it) that did not co-occur directly in training, but can be connected through things that did co-occur.

**Strengths:**

This paper is written in an exceptionally clear way, it's a pleasure to read. Additionally, it's one of those rare papers that is able to convincingly connect a very theoretical, toy, controlled result to a real-world phenomenon that we, as of yet, understand poorly. I feel like I understand zero-shot CoT better after reading this paper, and am very excited about follow-up work in this direction.

Other strengths:
- the authors have a simple theoretical result that helps shape intuitions before diving into the experimental section
- the authors convincingly apply control conditions to isolate the effect of the locality of the training experience, showing the same thing does not show up when there is no locality or the locality is "wrong" (clusters are from another DAG, meaning they are not actually local)
- the results are very clear; there is a reasoning gap for direction prediction versus reasoning when there is a locality in the variables encountered during training
- the authors additionally show benefits of this type of training data structure; data-efficiency
- the authors' conclusion is convincing: reasoning can help when a learner needs to connect concepts that have not directly been seen during training, but those concepts are connected through other concepts that have been seen together

**Weaknesses:**

My main point of weakness with this paper is that the hypothesised connection to actual zero-shot CoT reasoning in SotA LMs, and with it, actual reasoning done by humans is not described explicitly enough. The authors cite Chan et al. (2022) here to be similar to their work ("Data distributional properties drive emergent .."), but I found the connection between that toy setup and real language slightly more convincing because they use a Zipfian distribution (which the vocabulary in language also follows to trade-off the speaker and listener effort due to ambiguity). So my question here is; how exactly does your setup relate to real-world language; what kind of data might we find these local structures where variables are connected through intermediate variables but not directly co-occur; what do the variables refer to in language? Can you give an example? And if the example is topics, can you work that example out a bit more clearly? Don't get me wrong here, I find the connection convincing, but I think the paper can benefit from some explicit reasoning about the connection of the setup to real-world language. In a sense, the setup in the paper works because the model can just generate variables randomly until it encounters the target variable, whereas in real-world reasoning arguably the intermediate reasoning steps are connected through some high-level or abstract similarity.

Relatedly, it seems like a very important factor for reasoning like presented in the paper here to work is the overlapping clusters; what happens if not every cluster overlaps, or if overlapping variables are dropped out more often? Do you think people can make "reasoning jumps" through language between clusters of variables that do not overlap? To put it differently; perhaps when humans (and language models) do not observe some variable as both a part of cluster 1 and cluster 2, they can still "jump" between those clusters due to some abstract similarity between the two (e.g. a latent variable connection instead of an actual overlapping cluster).

**Questions:**

Most important questions are listed in weaknesses. Other questions that are less important here:
- Should line 138 say it only assigns non-zero probability to **adjacent** pars and not **non-adjacent** pairs as stated now?
- I think the first control condition can benefit from some more explanation; are the values still from the right Bayes net and only the local structure is wrong? I.e. is it for example a Bayes net like in Figure 1A but the drawn lines of clusters are not actually local clusters, but the arrows and conditional probabilities are the same?
- Random remark: Negative scaffolding intuitively seems to map onto this idea from LLM literature where people tried to test the hypothesis that CoT reasoning only works because it allows the model to output extra tokens. Tested with a control condition where they generate random tokens and then ask for the answer and then show that this works worse than CoT, meaning that the actual information the model outputs in its reasoning traces is important for performance.



**Limitations:**

The limitations are adequately addressed, but I think the authors should be more explicit about about how well this setup maps onto real language models and reasoning, focusing also on the requirement of overlapping clusters, and where the setup simplified things compared to real-world situations.

---

> ### Author Rebuttal · Authors · 2023-08-09
>
> We thank the reviewer for the thoughtful and supportive review. We are glad that the reviewer found our paper to be a pleasure to read and that they feel they understand zero-shot chain-of-thought better having read it.
>
> We have responded to the main weakness the reviewer identified, about the connection to actual CoT reasoning, in the main author rebuttal, but we also provide more specific responses below, along with responses to other points the reviewer brought up.
>
> > So my question here is; how exactly does your setup relate to real-world language; what kind of data might we find these local structures where variables are connected through intermediate variables but not directly co-occur; what do the variables refer to in language? Can you give an example? And if the example is topics, can you work that example out a bit more clearly? Don't get me wrong here, I find the connection convincing, but I think the paper can benefit from some explicit reasoning about the connection of the setup to real-world language.
>
> We will revise the paper to include more discussion of the connection between real-world reasoning and our setting. We will consider a detailed example to make the effect of topic structure clearer. For example, Wikipedia articles tend to mention the capital cities of countries and the climate of cities, but few articles directly talk about the climate of a country’s capital. If we were to ask a model trained on Wikipedia “What is the climate of France’s capital?” it would likely fail to answer directly. However, “France,” “capital,” and “Paris” co-occur frequently in the training set and “Paris” and “Oceanic climate” co-occur. By working through the intermediate reasoning step “Paris is the capital of France”, the language model should be able to answer the question correctly. It might produce a chain of thought like “The capital of France is Paris. The climate of Paris is oceanic. So the climate of France’s capital is oceanic.”
>
> > In a sense, the setup in the paper works because the model can just generate variables randomly until it encounters the target variable, whereas in real-world reasoning arguably the intermediate reasoning steps are connected through some high-level or abstract similarity.
>
> We agree with this point: locality is probably not the only factor making reasoning useful in natural language. Seeing examples of reasoning traces in the training corpus might lead the model to learn reasoning strategies that make use of high-level similarities between topics. Even more importantly, few-shot in-context-learning likely helps models generate relevant variables for CoT. (Here we studied only the 0-shot case.) This is an interesting direction for future research.
>
> > Relatedly, it seems like a very important factor for reasoning like presented in the paper here to work is the overlapping clusters; what happens if not every cluster overlaps, or if overlapping variables are dropped out more often? Do you think people can make "reasoning jumps" through language between clusters of variables that do not overlap? To put it differently; perhaps when humans (and language models) do not observe some variable as both a part of cluster 1 and cluster 2, they can still "jump" between those clusters due to some abstract similarity between the two (e.g. a latent variable connection instead of an actual overlapping cluster).
>
> Yes, one major difference between natural language and our setting is that natural language allows us to make abstract statements which can connect concrete facts in novel and unexpected ways. For instance, statistical information about predicates may influence joint probabilities of a variety of propositions that include them. While the focus of this work was to find a minimal case in which outputting extra information between a “question” and “answer” helps to produce better answers, the question of how abstract knowledge might enable better reasoning is an interesting direction for future work. We will update the discussion to mention this direction.
>
> > I think the first control condition can benefit from some more explanation; are the values still from the right Bayes net and only the local structure is wrong?
>
> Yes, that is correct. The values of the variables are still taken from the right Bayes net, but the local neighborhoods are based on a different Bayes net with a different structure. We will make this clearer in the paper.
>
> > Random remark: Negative scaffolding intuitively seems to map onto this idea from LLM literature where people tried to test the hypothesis that CoT reasoning only works because it allows the model to output extra tokens. Tested with a control condition where they generate random tokens and then ask for the answer and then show that this works worse than CoT, meaning that the actual information the model outputs in its reasoning traces is important for performance.
>
> Great point! Simply outputting extra tokens is not helpful in our setting. Step-by-step reasoning only works when the steps are relevant to the prediction task at hand. We will make this connection in the revision.

---

> > ### Comment · Reviewer_jyTS · 2023-08-11
> > **Thanks for the rebuttal**
> >
> > Thanks for the rebuttal, my points are adequately addressed. I think this paper is going to be a valuable insight to the community, and I will argue for its acceptance, but I urge the authors to be more explicit in a final version about the conditions under which this type of intermediate variable generation will be useful (e.g. overlapping clusters) and the distinctions between this setup and natural language. I'll increase my score to an 8.

---

### Official Review · Reviewer_tGUi · 2023-07-06

**Soundness:** 3 good
**Presentation:** 3 good
**Contribution:** 3 good
**Rating:** 7
**Confidence:** 4

**Summary:**

The starting point of this paper is the observation that large language model benefit from chain-of-thought reasoning. Namely, when prompt with a reasoning task, LLMs benefit from generating intermediate steps before reaching the final answer. The paper investigates this phenomenon. The authors hypothesize that this is a result of local structures in the training data where variables that often appear together have strong influence on each other, thus a model can generate chains of local connections and by that obtain relations between remote variables that do often appear together in the training data. To explore this hypothesis, the authors train an LLM on samples from randomly generated Bayes nets and task the model with inferring the conditional probability of one variable in the bayes net given another. They show that when the model is trained on samples that only include a subset of local variables then generating a chain of conditional probabilities involving adjacent variables,
predicts the conditional probability involving two remote variables whith much higher accuracy than a model that is tasked with predicting the remote relation directly. They refer to this phenomenon as the “reasoning gap”.  In contrast, when the model is trained either on the entire set of variables, or on a subset of variables from an irrelevant locality, that reasoning gap vanishes. They also prove a theoretical guarantee in the special case when the Bayes net is a simple chain of variables.


**Strengths:**

The topic of chain-of-thought reasoning in LLMs has garnered a lot of attention in the community recently. It is important both from a theoretical as well as applicable standpoints to investigate the conditions under which chain-of-thought reasoning is useful. The authors suggest an interesting set of experiments to do that as well as some theoretical work. At least for the model that was used in the experiments, the results and the conclusions are convincing. The paper is well written.

**Weaknesses:**

The authors ran extensive experiments, but the evaluation uses only one LLM architecture. To draw conclusions about chain-of-thought reasoning in LLMs in general, I would expect a larger set of model architectures to be part of the experiments. For example, it would be interesting to understand how the performance is influenced by the model’s size.

**Questions:**

Line 139 - shouldn't it be "adjacent" instead of "non-adjacent"?
Theorem 3.1 - what is q^hat?

**Limitations:**

The authors adequately addressed the limitations.

---

> ### Author Rebuttal · Authors · 2023-08-09
>
> We thank the reviewer for raising the important issue of architecture choice. We also appreciate that the reviewer finds the topic of chain-of-thought reasoning important and finds our results convincing.
>
> We have responded to the reviewer’s point about the choice of architecture, including new results with different architectures, in the main author rebuttal. We respond to other questions below.
>
> > Line 139 - shouldn't it be "adjacent" instead of "non-adjacent"?
>
> Yes, this was a typo. Thank you for catching it.
>
> > Theorem 3.1 - what is q^hat?
>
> $\hat{q}$ refers to estimators that are constructed from the raw predictions of the risk minimizer $q$. $\hat{q}$ alone is not defined, but $\hat{q}_D$ and $\hat{q}_S$ are estimators we define. See Section 2.2 for definitions of these different estimators.

---

> > ### Comment · Reviewer_tGUi · 2023-08-11
> > **My main concern has been addressed**
> >
> > I thank the authors for the thoughtful comments and for addressing my main concern. Now that it has been lifted, I'll be glad to see this paper presented at the conference, so I have raised my rating.

---

### Official Review · Reviewer_fiMT · 2023-07-07

**Soundness:** 3 good
**Presentation:** 2 fair
**Contribution:** 3 good
**Rating:** 7
**Confidence:** 3

**Summary:**

This paper provides a theoretical analysis of situations in which chain-of-thought reasoning should be helpful. They do this by considering the task of predicting variable values in a Bayes net. Specifically, it is a Bayes net where child nodes are a nearly-deterministic function of their parents. During training, the network sees sets of nodes and is asked to predict the value of a target node. At evaluation time, the network is asked to predict a held-out target node based on another held-out node.

They compare three approaches: (a) direct prediction, where the target node is predicted immediately (b) scaffolded prediction, where the model is prompted to predict all the nodes between the source node and target node, and (c) free prediction, where the model chooses itself which intermediate nodes to generate.

They find that in Bayes nets where relevant local variables are observed near each other, scaffolded prediction and free generation outperform direct prediction. If the observations shown are not relevant to the task, all approaches do poorly. If observations are not locally structured, scaffolded generation does well, but we don't see a benefit from free generation.

They show theoretically that in this setting "chain-of-thought" (i.e. scaffolded generation) produce better predictions and back this up with empirical tests.

**Strengths:**

Originality/significance: I am not familiar with theory in this area, but this appears to be a new and exciting result. It could be useful for the field by providing people with intuition about what types of datasets are promising candidates to use with chain of thought.
Quality/clarity: They present theory and complementary empirical analysis. They consider a few different forms of reasoning and a few different types of observation frequency structures to confirm that their hypotheses hold across these conditions.

**Weaknesses:**

* It took a long time to understand the intuition behind how the theoretical and empirical results relate to the claims about reasoning in sequence models. Possibly this could be made clearer?
* I wish there was an analysis of how accurate the intermediate reasoning steps were in the different conditions (scaffolded vs free generation).
* The experiments look at conditions where each variable's value is nearly deterministic based on its parents. I wish there was either an analysis of what happens when this is not true or a discussion of whether the types of natural language tasks where chain of thought is most helpful share this nearly-deterministic property.
* The experiments average over 10 Monte Carlo rollouts for the scaffolded and free generation cases. I wish there was an analysis of how chain of thought compares to direct prediction in the case of a single rollout, which is often the case being considered in language models.

**Questions:**

In Section 3, line 189, should "$p_{obs}$ only assigns non-zero probability to non-adjacent variable pairs" say "adjacent" not "non-adjacent"?

I'm not sure I fully understood the significance and takeaways of the analysis and experiments. I've summarized my understanding below, but please correct me if I'm misunderstanding:

In the theoretical analysis, my intuitive understanding of the result is that if your training data consists of only adjacent pairs, then when you try to evaluate on non-adjacent pairs a perfect estimator will be very wrong (since those samples are never seen, so it will default to a uniform prior).  On the other hand, if you predict probabilities of non-adjacent pairs by chaining a set of probabilities for adjacent pairs, then each term in the chain is in-distribution and you’ll get a better estimate.  Is that the correct intuition? If so, (a) is there any way to make the intuition behind the result clearer, and (b) could you maybe make a higher analogy with language (e.g. explaining what a “chain” of variables would look like in language, what it means for training data to consist mostly of adjacent pairs, etc.)

It seems like reasoning working well consists of two thing: (A) the model must be able to produce a reasoning trace which is relevant to the task at hand, and (B) the model must be able to use the reasoning to produce a better final answer.
* It seems like the “scaffolded” condition checks if (B) is true, and the “free generation” condition checks if both (A) and (B) are true. The theoretical results suggest that (B) should is true but don’t touch on (A). Is there any theoretical reason to think (A) should be true primarily for locally-structured data? Or is the intuition just that locally-structured observations will bias the model towards generating intermediate variables easy to predict accurately b/c the model is conditioning on local context and are relevant to the task at hand?


**Limitations:**

Limitations are adequately addressed.

---

> ### Author Rebuttal · Authors · 2023-08-09
>
> We thank the reviewer for their detailed review. We appreciate that the reviewer found the paper’s result new and exciting, and that they thought it provides useful intuition about when chain-of-thought is useful.
>
> We respond to the weakness about variable values being nearly deterministic given parents in the general author response, as well as the comment about the experiments averaging over 10 Monte Carlo rollouts. We also respond to the remaining questions below.
>
> > In the theoretical analysis, my intuitive understanding of the result is that if your training data consists of only adjacent pairs, then when you try to evaluate on non-adjacent pairs a perfect estimator will be very wrong (since those samples are never seen, so it will default to a uniform prior). On the other hand, if you predict probabilities of non-adjacent pairs by chaining a set of probabilities for adjacent pairs, then each term in the chain is in-distribution and you’ll get a better estimate. Is that the correct intuition? If so, (a) is there any way to make the intuition behind the result clearer, and (b) could you maybe make a higher analogy with language (e.g. explaining what a “chain” of variables would look like in language, what it means for training data to consist mostly of adjacent pairs, etc.)
>
> Yes, that is exactly the right intuition. The perfect estimator, with respect to the risk defined in the paper, will default to the uniform prior for non-adjacent pairs. When we chain the risk minimizer’s estimates of adjacent pairs, we indeed make use of “in-distribution” pairs and therefore can reduce bias.
>
> Regarding improving the clarity of the theoretical result, in the revised version of the paper, we can include a figure that illustrates the theoretical formulation more carefully.
>
> > It seems like reasoning working well consists of two thing: (A) the model must be able to produce a reasoning trace which is relevant to the task at hand, and (B) the model must be able to use the reasoning to produce a better final answer.
>
> We do not have concrete theoretical results for (A). The intuition behind why (A) happens in practice in free generation is that the relevant variables to reason through tend to be close to the observed variable and the training set consists of local clusters. In practice, free generation generates several variables near the observed variable, only some of which are relevant but which are sufficiently relevant in aggregate. It might be possible to prove analogous theoretical results for (A) but we leave a thorough characterization of free generation for future work.

---

> > ### Comment · Reviewer_fiMT · 2023-08-16
> > **Concerns addressed!**
> >
> > Raising to 7.

---

### Official Review · Reviewer_jbx9 · 2023-07-19

**Soundness:** 4 excellent
**Presentation:** 4 excellent
**Contribution:** 4 excellent
**Rating:** 7
**Confidence:** 3

**Summary:**

This work investigates why and how chain-of-thought reasoning works in language models in the aspect of
**local structure** in the training data. To this end, this work first proves the hypothesis that there exists a reasoning gap where reasoning through intermediate variables improves inference and then tests the hypothesis by training an autoregressive language model on samples from Bayes nets but only including
a subset of variables in each sample, considering the estimators including direct prediction, scaffolded generation, and free generation. The key findings are intermediate steps are only helpful when the training data is locally structured with respect to dependencies between variables and that the combination of locally structured observations and reasoning is much more data-efficient than training on all variables.

This work begins with the human practice of step-by-step reasoning and reviews the recent progress of a similar mechanism --- the intriguing chain-of-thought reasoning in large language models. Then the paper asks the question of why step-by-step reasoning helps, which may not only help understand how the large language models work but also provide insight into the origins of human reasoning.

The basic hypothesis is that chain-of-thought reasoning is useful in language models due to the local structure in the training data. Such a hypothesis is intuitive as human reasoning transcends local bounds, supporting plans and conclusions that span in time and space. The meaning of local structure may be interpreted as observations occurring in overlapping neighborhoods of concepts.

To verify the hypothesis, the authors conduct theoretical analysis and empirical experiments, finding that performing conditional inference by first generating intermediate variables improves the ability of a language model to match true conditional probabilities only when the training data is structured locally with respect to strong dependencies and the intermediate variables are relevant to the relationship between the variables of interest. Finally, the work also provides insights into three aspects: (i) when reasoning helps --- the observation distribution has the correct locality structure; (ii) when reasoning is unnecessary -- the observed and target variables co-occur in the training distribution; (iii) When reasoning fails --- data with the wrong locality structure.

Overall, this work studies an important and timing research topic towards why and when step-by-step works in language models. This work provides comprehensive theoretical and experimental results to support the hypothesis. This work also provides useful insights into solving reasoning tasks, as well as dataset construction to amplify the capacity of large language models to perform step-by-step reasoning.

**Strengths:**

1. The topic studied in this work is very important and has attracted increasing interest in the community. This kind of work is useful to advance our scientific understanding of why the intriguing chain-of-thought reasoning works (or fails on some tasks), and helps facilitate future studies on training dataset construction and prompting techniques to amplify the capacity of large language models to solve complex reasoning tasks.

2. The theoretical part clearly formulates the problem. It also introduces effective approaches for estimation, followed by convincing experiments on a real-world language model (though with the smaller-scale GPT-2 instead of the real large language models).

3. Many great insights can be found in the paper, after taking into account the influence of data complexity, when reasoning works, becomes unnecessary, and even fails. The findings basically align the real-world practice when applying chain-of-thought prompting in different reasoning tasks.

**Weaknesses:**

1. The paper can be improved by taking large language models as the backbone and verifying the hypothesis on real-world datasets where chain-of-thought prompting techniques are widely applied. It would be interesting if this work could provide effective ways to identify the locality of a dataset, which may answer the commonly found yet unresolved question of why chain-of-thought techniques work quite well in arithmetic tasks but fail at some standard natural language understanding tasks like simple classification.

2. The connection and the difference with existing studies can be further clarified. In the introduction part, this work mentioned another jargon --- "burstiness" which is confusing. It is not clear how burstiness and locality differ from each other. Besides, the motivation for choosing locality as the research topic is not clear, either. It is interesting to see more elaborations on how the hypothesis is derived.

**Questions:**

Why does this work choose a smaller version of GPT-2 for experiments instead of a large language model? Scaling laws affect the step-by-step reasoning ability of language models. Small models may commonly fail at step-by-step reasoning compared with direct reasoning. In contrast, large language models may be the better backbone to verify the hypothesis of this work.

**Limitations:**

The authors have provided thoughtful descriptions of the limitations.

---

> ### Author Rebuttal · Authors · 2023-08-09
>
> We thank the reviewer for the detailed and thoughtful review, and are happy to see that the reviewer found our theoretical analysis and simulation results insightful.
>
> We describe how we will address the second weakness in the general author rebuttal. We also respond to the question about the choice of architecture by showing that a similar reasoning gap appears for multiple different architectures.

---

> > ### Comment · Reviewer_jbx9 · 2023-08-11
> > **Thanks for the rebuttal.**
> >
> > I appreciate the authors' clarifications. My concerns have been well addressed.

---

### Author Rebuttal · Authors · 2023-08-09

We thank the reviewers for their thoughtful comments on this paper. These comments have informed additional analyses and clarifications.

# Architecture
Reviewers Jbx9 and tGUI ask about our choice of architecture. Jbx9 asks why we chose a smaller model given that models smaller than GPT-3 often fail at step-by-step reasoning, while reviewer tGUi identifies our use of only one architecture as a weakness.

To these points, we first point out that our theoretical analysis applies to any autoregressive empirical risk minimizer. We prove that a reasoning gap should exist between direct prediction and scaffolded generation regardless of the specific architecture (for simple world distributions). We can interpret our learning curve results as evaluating whether a reasoning gap exists at different levels of empirical risk, or equivalently perplexity. The simulated data we trained transformers on is much simpler than natural language, so it is expected that a smaller model is capable of learning the data well enough for reasoning to be beneficial. Figure 1 in the PDF shows the mean perplexity on a validation dataset across Bayes nets vs. the size of the reasoning gap for each checkpoint in the geometrically-sized local neighborhood model. These results show that a model must achieve a certain perplexity before the reasoning gap emerges, but it exists for a range of perplexities achievable by our architecture.

We have also run additional simulations in which we train different transformer architectures on geometrically-sized local neighborhoods from the same Bayes nets. We use a smaller model (4,473,600 parameters), a larger model (86,628,864 parameters, the base gpt-2 architecture) and model with a similar number of parameters, but larger embeddings and fewer layers (39,887,872 parameters). We also use a tiny model which is too small to learn the distribution of the data (8,644 parameters). All models except for the tiny model exhibit similar reasoning gaps, suggesting that our findings are not specific to one architecture. Performance across architectures and estimators is reported in Figure 2 of the PDF, which we will include in the appendix of the paper.

# Number of samples
Reviewer fiMT was interested in seeing results with a single Monte Carlo sample used in the scaffolded and free generation estimators. We have run this analysis for all numbers of samples between 1 and 10 and reported the results in Figure 3 of the PDF.

While we used the mean squared error as a practical measure of the error of a language model, we could decompose MSE into bias and variance. The free and scaffolded generation estimators have lower bias, but higher variance, than direct prediction. Free and scaffolded generation have higher MSE compared to direct prediction with one sample, but averaging over several samples (at least 3 in our setting) leads them to have lower MSE than direct prediction. This finding has bearing on when we should expect generating multiple chains of thought, as is done in confidence methods (e.g. Wang et al., 2022), to be valuable. The specifics of the bias-variance trade-off depend on the underlying stochasticity of the Bayes net, so different environments might call for different numbers of samples. We will discuss this result in the paper.

# Connection to real-world reasoning
Reviewers Jbx9, fiMT, and jyTS mentioned that the connection between our setting and real-world reasoning in humans and language models was not clear in the paper. We are ultimately interested in studying the improvement at answering questions resulting from generating intermediate information between a question and its answer. Our setting of inference for held-out pairs of variables in a Bayes net is a minimal case where this happens. For example, in a simple Bayes net with three variables, A: it rained last night, B: the grass is wet, and C: mowing the lawn will be difficult, reasoning through the intermediate variable B might be necessary if A and C have not been encountered together directly in training data. We will revise the introduction to make the connection clearer.

Reviewer fiMT asks about our choice to favor strong dependencies in generating Bayes nets. We chose to generate data this way to ensure that there are non-adjacent pairs of variables with high mutual information. If we sampled probabilities uniformly, mutual information would decay rapidly with distance and conditional probabilities for held-out pairs would be almost identical to marginal probabilities. Still, our Bayes nets have considerable randomness as the Beta distribution we use generates conditional probabilities between 0.1 and 0.9 32.7% of the time. We expect reasoning to be most useful in environments with strong dependencies, like in math word problems where truth values of statements are deterministic. An alternative way of looking at this is that strong long-range dependencies are themselves a precondition for reasoning – otherwise using the marginal frequency of the conclusion is enough.

# Clarifications
We will clarify the motivation behind our choice to study locality and how it differs from related ideas like burstiness. While burstiness concerns the distribution of a single class over time, locality is about which classes co-occur with each other. Co-occurrence patterns are relevant because reasoning connects different concepts.

We also thank reviewers fiMT and tGUi for identifying typos and unclear points in the paper. In particular, we have fixed the typo on line 138/139 where “non-adjacent” should be “adjacent”.

We once again thank the reviewers for taking the time to give detailed feedback on this work. Their comments have substantially improved this paper.

# References
Wang, X., Wei, J., Schuurmans, D., Le, Q. V., Chi, E. H., Narang, S., ... & Zhou, D. (2022, September). Self-Consistency Improves Chain of Thought Reasoning in Language Models. In The Eleventh International Conference on Learning Representations.

---

### Decision · Program_Chairs · 2023-09-21

**Decision:**

Accept (oral)

**Comment:**

This paper aims to answer a very fundamental question -- in what situation chain-of-thought reasoning should be helpful. It explores this problem by considering the task of predicting variable values in a Bayes net, where child nodes are a nearly-deterministic function of their parents. The task is giving the values of a set of nodes and predicting the value of a target node.
It also compares three approaches: (a) direct prediction, where the target node is predicted immediately (b) scaffolded prediction, where the model is prompted to predict all the nodes between the source node and target node, and (c) free prediction, where the model chooses itself which intermediate nodes to generate.

The result shows that
1. if relevant local variables are observed near each other, scaffolded prediction and free generation outperform direct prediction.
2. If the observations shown are not relevant to the task, all approaches do poorly.
3. If observations are not locally structured, scaffolded generation does well, but there is no benefit from free generation. Theoretically analysis shows that in this setting "chain-of-thought" (i.e. scaffolded generation) produces better predictions and empirical tests also confirms.


Strengths:
1. The result is significant in providing people with intuition about what types of datasets are promising candidates to use with chain of thought.
2. The approach is novel, and the theory and complementary empirical analysis are solid.

Weaknesses:
1. lack of a practical way to measure the locality of a dataset
2. the experiment is based on the case where each variable's value is nearly deterministic based on its parents. It is not clear how that generalizes to real datasets (e.g. language tasks)
3. while theoretical arguments can often be made with smaller model sizes, the connection of these explanations to observed phenomenon on larger models seems like it would require actually experimenting with larger models.